# Posterior Cruciate Buckling Angle Variations Are Associated with Different Patterns of Medial Meniscus Tears in Anterior-Cruciate-Deficient Knees: Results of a Prospective Comparative Magnetic Imaging Resonance Study

**DOI:** 10.3390/healthcare12161553

**Published:** 2024-08-06

**Authors:** Simone Cerciello, Michele Mercurio, Katia Corona, Lorenzo Proietti, Giovanni Di Vico, Matthew Charles Giordano, Brent Joseph Morris

**Affiliations:** 1Department of Life Sciences, Health and Health Professions, Link Campus University, 00165 Rome, Italy; simone.cerciello@me.com; 2Department of Orthopedic and Trauma Surgery, “Magna Græcia” University, “Mater Domini” University Hospital, V.le Europa, 88100 Catanzaro, Italy; mercuriomi@gmail.com; 3Department of Medicine and Health Sciences “Vincenzo Tiberio”, University of Molise, Via Giovanni Paolo II, 86100 Campobasso, Italy; 4Casa di Cura Villa Betania, 00195 Rome, Italy; proiettilorenzo@hotmail.com (L.P.); matgiordano@hotmail.com (M.C.G.); 5Department of Orthopaedic and Trauma Surgery, Casa di Cura San Michele, 81024 Maddaloni, Italy; divicogiovanni@gmail.com; 6Baptist Health Medical Group Orthopedics and Sports Medicine, Lexington, KY 40503, USA; brent.joseph.morris@gmail.com

**Keywords:** ACL tear, PCL buckling angle, magnetic resonance imaging, interobserver reliability, intraobserver reliability

## Abstract

Background: The diagnosis of anterior cruciate ligament (ACL) tear relies on clinical evaluation and magnetic resonance imaging (MRI). Direct and indirect signs of ACL tear have been described with MRI evaluation. Posterior cruciate ligament (PCL) buckling has been described as an indirect radiographic sign of an ACL tear. Purpose: The aim of the present study was to assess the variations in PCL buckling angles in patients with ACL tears and in patients with isolated lesions in the posterior horn of the medial meniscus. In addition, the influence of different patterns of medial meniscus tears in ACL-deficient knees was investigated. Finally, the influences of risk factors such as tibial slope, delay from injury to surgery, absence of medial meniscus tear, degree of Lachman and pivot shift testing were also assessed. Study design: This was a cohort study. Methods: A total of 154 patients (78 in the group with ACL tear and 76 in the control group) were assessed with MRI and lateral weight-bearing X-ray to assess PCL buckling angle and tibial slope by two independent observers. The presence of a medial meniscus bucket handle or ramp lesion of the medial meniscus was assessed and recorded at the time of surgery. Results: PCL buckling angle measurement was highly reliable, with an ICC of 0.866 and 0.894, respectively, in the study group and the control group for interobserver reliability. The intrarater reliability was found to be high in PCL buckling angle for the study group [ICC = 0.955] and the control group [ICC = 0.943]. The mean angle in patients with ACL tear was 110.7 ± 15.2° and 115.3 ± 16.2° (for the two examiners) and 111.4 ± 12° and 114 ± 14.5° (for the two examiners) in patients with an intact, healthy ACL. An association emerged between bucket handle tears of the medial meniscus (*p* = 0.010) and a decreased PCL buckling angle and between ramp lesions of the medial meniscus and increased PCL buckling angle both (*p* = 0.024). Conclusions: Good inter- and intraobserver reliability for the measurement of the PCL buckling angle was observed. Increased PCL buckling angle values were observed in patients with concomitant ACL and bucket handle tears of the medial meniscus, while decreased angle values were observed in those who had ACL tear and ramp lesion of the medial meniscus. No statistically significant difference in the PCL buckling angle emerged between patients with ACL tears and those who had a healthy, intact ACL.

## 1. Introduction

Anterior cruciate ligament (ACL) tears are common injuries, with an annual rate of 68.6 per 100,000 person-years [1]. The diagnosis of ACL tears is based on both clinical examination and imaging. Clinical examination relies on patient history (episodes of instability following a knee sprain) and specific clinical tests (anterior drawer, Lachman, and pivot shift tests).

Magnetic resonance imaging (MRI) is considered the study of choice for assessing the status of the ACL and detecting its tears [2,3]. A recent meta-analysis showed that MRI has a sensitivity (SE) of 87%, a specificity (SP) of 90%, a positive likelihood ratio (LR+) of 6.78, a negative likelihood ratio (LR-) of 0.16, and a diagnostic odds ratio (DOR) of 44.70 in diagnosing ACL tears [4]. The diagnosis of ACL tear on MRI is usually based on direct and indirect radiologic signs [5,6]. Direct signs include fiber discontinuity, swelling, increased signal on T2 or fat-saturated PD, fiber discontinuity, abnormal anterior cruciate ligament orientation relative to the intercondylar (Blumensaat) line, and empty notch sign. Indirect signs are bone bruising [7] or posterior cruciate ligament (PCL) buckling (angulation of the PCL) [8].

PCL buckling results from anterior translation of the tibia in ACL deficiency and can be observed in cases of acute or chronic ACL tears. The PCL buckling angle is a line drawn through the center of the proximal and distal PCL [9], and it is considered abnormal if the angle is less than 105° [8]. McCauley et al. observed that a PCL angle of less than 105° had sensitivities of 72% and 74%, with corresponding specificities of 79% and 86% for ACL tears [7]. Gentili et al. showed that a PCL buckling angle of less than 107° had a sensitivity of 52% and specificity of 94% in diagnosing ACL tears [10,11]. Some studies associated a sigmoid or curved appearance of the PCL more with chronic than acute ACL tears [11,12].

The primary purpose of the present study was to measure the PCL buckling angle in patients with intact ACLs and in patients with torn ACLs. The secondary aim was to assess the influence of concomitant different meniscus tear patterns (posterior horn tears, bucket handle, and ramp lesions) in ACL-deficient knees on PCL buckling. In addition, the influence of posterior tibial slope, which is a risk factor for increased anterior tibial translation, on PCL buckling was investigated. Finally, we sought to assess the reproducibility of the measurement of the PCL buckling angle. The hypothesis was that the PCL buckling angle, which is calculated from an MRI sagittal scan with the knee in full extension laying on the MRI table, would not be precise in assessing the static anterior drawer, which is the cause of PCL bending. Other factors such as bucket handle of the medial meniscus or medial meniscus ramp lesions would have had more influence in producing such a static effect.

## 2. Materials and Methods

The present study was designed as a prospective study with a control group. It was based on electronic charts (clinical, MRI, and X-ray).

Seventy-eight patients undergoing ACL reconstruction from 2019 to 2020 from a single center were prospectively enrolled as the study group and informed consent was obtained. All patients had a clinical and MRI diagnosis of ACL tear, which was confirmed at the time of the arthroscopic procedure. The delay between injury and surgery was recorded, and its influence on the PCL angle was assessed. Exclusion criteria included previous surgeries of the affected knee, previous or concomitant lesions of the PCL and of the medial and lateral collateral ligaments (MCL and LCL), and previous fractures of the ipsilateral femur and tibia.

All patients were clinically evaluated in the surgical ward by a senior surgeon trained in sports medicine. The degree of anterior–posterior instability was recorded with Lachman testing and graded according to the original description by Adler et al. [13], with Grade 1 (mild): 3–5 mm translation of the tibia on the femur, Grade 2 (moderate): 5–10 mm translation of the tibia on the femur, and Grade 3 (severe): >10 mm translation of the tibia on the femur.

Rotatory instability was evaluated with the pivot shift test. It was graded according to the classification by Jacob et al. [14], with Grade 1 being abnormal movement when the leg was held in a neutral position and Grade 3 being when abnormal movement was observed when the leg was held in external rotation. The integrity of the MCL and LCL was confirmed with the varus and valgus stress test. MRI as well as X-rays (including weight-bearing lateral view with the knee at 15° flexion) of the involved knee were available for all patients and were performed within 20 days prior to surgery. All imaging measurements were carried out on digital X-ray using a DICOM medical image viewer (Horos Project; Purview, Annapolis, MD, USA). On the lateral view of the X-ray, the posterior tibial slope was measured according to the method described by Dejour et al. [15]. The proximal anatomic axis of the tibia was first drawn by connecting the midcortical diameters of the tibia 5 and 10 cm distal to the joint line. A reference line was created perpendicular to this anatomic axis. Tibial slope was defined as the angle between the reference line, and a line was drawn tangent to the uppermost anterior and posterior edges of the medial tibial plateau (Figure 1).

On the sagittal view of the MRI, the angle formed between the proximal and distal parts of the PCL was measured, evaluating the same sagittal cut according to the method described by Yoon et al. [9]. The angle is formed by the intersection of two lines, which follow the proximal and distal parts of the PCL (Figure 2).

All measures were carried out by two independent surgeons blinded with the aim of the study to assess the interobserver reliability. The same surgeons performed the same imaging measurements after four weeks to assess the intra-observer reliability. During arthroscopy, associated lesions of the menisci were assessed and recorded to assess their influence on the PCL angle.

Additionally, in the same period, 76 patients undergoing a knee arthroscopy for isolated tears of the posterior horn of the medial meniscus and intact ACL were enrolled as a control group. Exclusion criteria were the same as for the study group. All patients were assessed with the same clinical and imaging protocol as the study group. The patients in the two groups were similar with regard to age, sex, and body weight.

### Statistical Analysis

Statistical analyses were performed using Statistical Package for Social Sciences (SPSS) version 24.0 (IBM-SPSS, New York, NY, USA). Data were tested for normal distribution by use of the Kolmogorov–Smirnov Z test. Continuous variables are expressed as frequencies (percentages). Statistical differences between measurements of the PCL buckling angle and the tibial slope conducted by the two independent reviewers and between the study and control group of each reviewer were tested by unpaired Student’s *t*-test. Inter- and intraobserver agreements for PCL buckling angle and tibial slope were evaluated by the intraclass correlation coefficient (ICC) 2-by-2 with a 95% confidence interval. The power of ICC values was interpreted according to the Landis and Koch classification [10] as follows: no agreement to slight agreement, <0.20; fair agreement, 0.21 to 0.40; moderate agreement, 0.41 to 0.60; substantial agreement, 0.61 to 0.80; and almost perfect agreement, 0.81 to 1.00.

A univariate logistic analysis was used to evaluate the relationship between each categorical variable and the PCL buckling angle. The variables that were noted to be significant for PCL buckling after univariate analysis were included in the multivariate linear regression analysis model, and the independent predictors of PCL buckling were finally determined. The level of significance was set at *p* < 0.05.

A prior sample size calculation was performed considering previous studies (G*Power 3 Software, (version 3.1.9.2, Institut fur Experimentelle Psychologie, Heinrich Heine Universitat, Dusseldorf, Germany)) [7,10]. A minimum of 68 patients per group was determined to satisfy medium–large effect size (Cohen’s *ƒ* = 0.33) with 80% power and a statistical significance at an alpha level of 0.05.

## 3. Results

The patient characteristics are described in Table 1 and Table 2. The groups were comparable in terms of age, male/female ratio, and the affected side.

No statistically significant differences in terms of the PCL buckling angle emerged between the groups for either the first or the second radiographic reviewer. The inter- and intraobserver reliability is summarized in Table 3. Tibial slope values were significantly higher in the study group compared to the control group (*p* = 0.007 and *p* = 0.001 for the two reviewers).

The interobserver reliability between the two radiographic reviewers showed strong agreement in PCL buckling angle evaluations in the study [ICC = 0.866] and the control [ICC = 0.894] groups, demonstrating that the data were reproducible. A substantial interobserver reliability was observed in tibial slope evaluations in the study [ICC = 0.787] and the control [ICC = 0.849] groups. The intrarater reliability between the same observer measurement was found to be high in the PCL buckling angle for the study group [ICC = 0.955] and the control group [ICC = 0.943]. A substantial intrarater reliability was observed in tibial slope evaluations in the study group [ICC = 0.669] and the control group [ICC = 0.698].

Univariate analysis showed a statistically significant correlation between PCL buckling angle and the following variables: ramp lesion (*p* = 0.047) and bucket handle tear (*p*= 0.004). No significant correlation was found between PCL buckling angle and Lachman Test (*p* = n.s), pivot shift test (*p* = n.s.), tibial slope (*p* = n.s.), medial meniscus tear (*p* = n.s.), no medial meniscus tear (*p* = n.s), and the delay between injury and surgery (*p* = n.s.). (Table 4)**.**

The following multivariate linear regression analysis showed a statistically significant difference between the PCL buckling angle, the ramp lesion (*p* = 0.024) and the bucket-handle tear (*p* = 0.010). (Table 5) In patients with ramp lesion, the PCL buckling angle (*p* = 0.024) was statistically superior in the control group, while it was statistically inferior in patients with bucket-handle tears of the medial meniscus (*p* = 0.010).

## 4. Discussion

The most important finding of the present study was the statistically significant association that emerged between bucket handle tears of the medial meniscus and smaller PCL buckling angles (104 ± 11°) (*p* = 0.010) and between ramp lesions of the medial meniscus and larger PCL buckling angles (114 ± 14°) (*p* = 0.024).

Conversely, no statistically significant difference in the PCL buckling angle emerged between patients with ACL tear (110.7 ± 15.2° and 115.3 ± 16.2° for the two reviewers) and patients without ACL tear (111.4 ± 12° and 114 ± 14.5° for the two reviewers). In addition, no correlation emerged between PCL buckling angles and the majority of the included risk factors. Tibial slope did not influence the degree of the PCL buckling angle (*p* = 0.378); however, it is a confirmed risk factor for ACL tear. In fact, a statistically significant difference was observed between the study group and the control group for reviewer 1 (9.3 ± 2.5° and 8.2 ± 1.8°; *p* = 0.007) and reviewer 2 (9.3 ± 2.8° and 7.7 ± 2.3°; *p* = 0.001). A longer delay between injury and surgery did not affect the degree of the PCL buckling angle (*p* = 0.138). Finally, measurement of the PCL buckling angle showed good interobserver and intraobserver reliability.

MRI is a reliable tool for confirming the diagnosis of ACL tears. Direct and indirect radiographic signs on MRI have been described. PCL buckling has been proposed as an indirect sign of ACL tear with cut-off values of 105° [7] and 107° [10]. In addition, PCL buckling has demonstrated 41% sensitivity, 70% specificity, 76% positive predictive value, and 35% negative predictive value for the diagnosis of full-thickness or partial-thickness tears of ACL reconstruction grafts [16]. The increased anterior tibial translation associated with ACL tears would be responsible for decreased PCL buckling angles. In this study, the PCL buckling angle was measured by two independent blinded reviewers both in patients with ACL tears (study group) and in those with isolated tears of the posterior horn of the medial meniscus (control group). No statistically significant differences emerged between the groups (SG and CG) tested by the same clinicians (clinician 1: 110.7 ± 15.2° and 111.4 ± 12°; *p* = 0.991) and (clinician 2: 115.3 ± 16.2° and 114 ± 14.5°; *p* = 0.421). This finding may be the consequence of the position of the knee during MRI. In supine positioning, the dynamic anterior tibial translation associated with ACL tear is reduced when compared to that in a weight-bearing situation. In such a condition it seems logical to have similar values between the two groups. The measuring method showed good intraobserver reliability (0.955 and 0.943 in the two groups). Ultimately, statistically significant differences emerged between the measurements performed by the two clinicians in the two groups (SG: 110.7 ± 15.2° and 115.3 ± 16.2°; *p* < 0.0001 and CG: 111.4 ± 12° and 114 ± 14.5°; *p* = 0.0035). The measuring method showed good interobserver reliability in the two groups, (0.866 and 0.894 respectively), though this was lower than the intraobserver reliability. This suggest that PCL buckling angle is an unreliable indirect radiographic sign of ACL tear.

The second notable finding is the clear correlation that emerged between a smaller PCL buckling angle and ramp lesions of the medial meniscus. It has been clearly demonstrated that the menisco-tibial attachment of the medial meniscus is a secondary stabilizer against anterior tibial translation in the case of an ACL injury with an intact medial collateral ligament [17]. In addition, it has been demonstrated that anterior tibial translation and external rotational laxity are significantly increased after the sectioning of the posteromedial menisco-capsular junction (equivalent to a ramp lesion) in an ACL-deficient knee [18]. It seems therefore logical for patients to experience an increased anterior tibial translation with a smaller PCL buckling angle in those with combined ACL and medial meniscus ramp lesions.

Conversely, a statistically significant correlation emerged between bucket handle tears of the medial meniscus and increased PCL buckling angle. This may have been due to the dislocation of the meniscus in the intercondylar notch, which may contact the PCL, causing PCL stretching and increasing the PCL buckling angle, and due to the slight knee flexion due to the bucket handle tear and to muscle contraction due to pain.

The last aspect that emerged from the present study was the absence of any correlation between PCL buckling angle and the majority of the examined factors, such as PCL buckling angle and tibial slope (*p* = 0.378). Although biomechanical studies reported that a higher tibial slope results in a greater anterior tibial translation, this seemed not to affect the PCL buckling angle [19,20]. On the contrary, tibial slope was significantly increased in patients with ACL tears compared to that in the control group both for observer 1 (9.3 ± 2.5° and 8.2 ± 1.8°; *p* = 0.007) and observer 2 (9.3 ± 2.8° and 7.7 ± 2.3°; *p* = 0.001), indicating that it is a risk factor for ACL tears. In addition, longer delays between injury and surgery, which are thought to stretch soft tissues increasing laxity, do not seem to influence the degree of the PCL buckling angle (*p* = 0.378).

The present study has several notable limitations. First of all, in the study group, only concomitant lesions of the medial meniscus were included. Including lateral meniscus lesions may have influenced the outcomes. Secondly, the degree of knee laxity (Lachman and pivot shift test) was only based on clinical evaluation. Thirdly, we noticed some inconsistencies among the measurements. Although high inter- and intraobserver reliability was observed, PCL buckling angles varied between the two observers by around 3° both in the study and the control group. Conversely, tibial slope measurements were similar among the two observers both in the study and control groups, with high interobserver reliability but much lower intraobserver reliability.

Although we acknowledge the limitation above, the present study has several notable findings. First of all, in ACL-deficient knees, a clear association between higher PCL buckling angle and bucket handle tear of the medial meniscus and between smaller PCL buckling angle and ramp lesions of the medial meniscus emerged. Secondly, the PCL buckling angle did not show a statistically significant difference between patients with or without ACL tears. Therefore, the angle should not be used as an indirect radiographic sign of ACL tearing. Thirdly, although an increased tibial slope did not seem to influence the PCL buckling angle, statistically increased values were observed in the cohort of patients with ACL tears. Fourthly, the methodology is reliable with high interobserver reliability.

## 5. Conclusions

Increased PCL buckling angle values were observed in patients with concomitant ACL and bucket handle tears of the medial meniscus, while decreased PCL buckling angle values were observed in those who had ACL tear and ramp lesion of the medial meniscus. This indicates the substantial influence of the peripheral structures on PCL tension. No statistically different PCL buckling angle values were observed between patients with ACL tears and in those who had a healthy, intact ACL. Increased tibial slope values were observed in the cohort of patients with ACL rupture, but it did not influence PCL angles.

## Figures and Tables

**Figure 1 healthcare-12-01553-f001:**
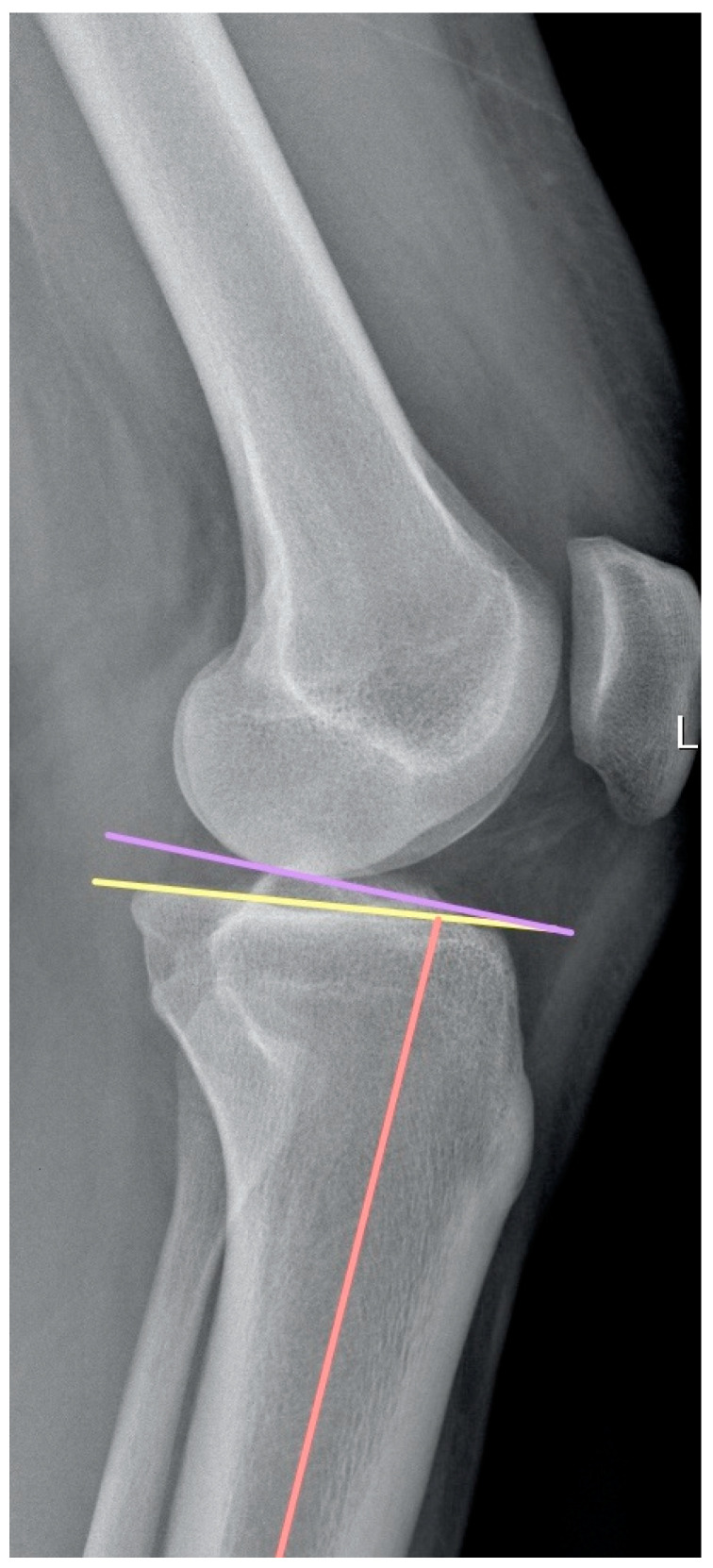
The posterior tibial slope was measured according to the method described by Dejour et al. [15] using the true sagittal view to measure the angle between the line perpendicular to the tibial diaphyseal axis (violet line) and the tangent to the most superior points at the anterior and posterior edges of the medial tibial plateau (yellow line).

**Figure 2 healthcare-12-01553-f002:**
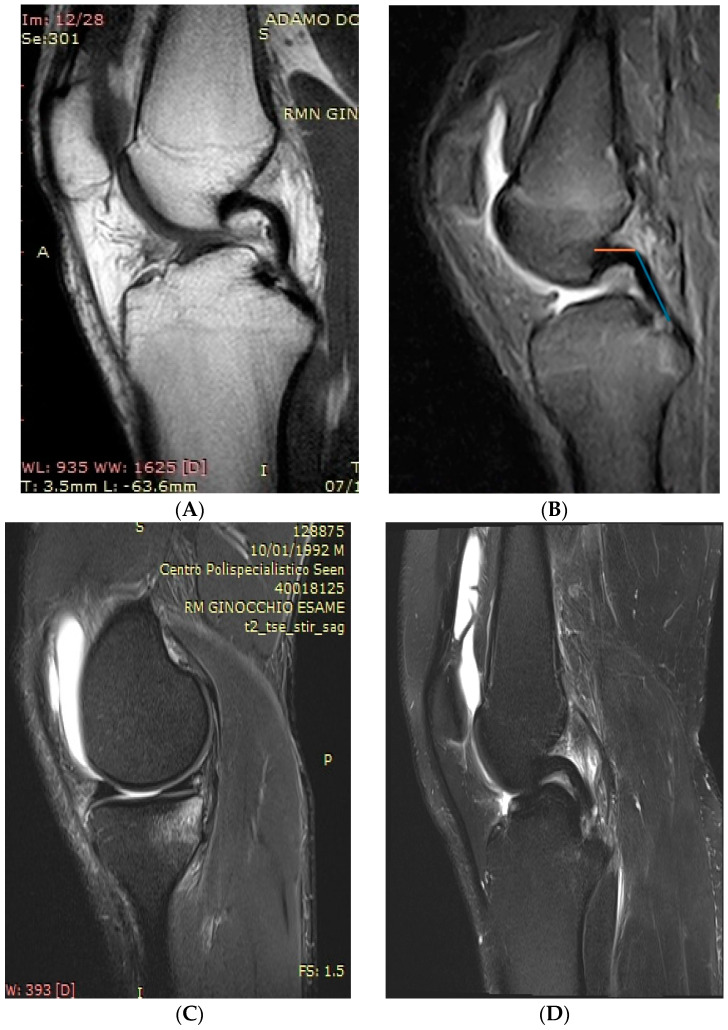
(**A**,**B**) PCL buckling is visible on sagittal views focusing on the center of the intercondylar notch. PCL buckling angle was calculated according to the method described by Yoon et al. [9]. It is formed by the intersection of two lines which follow the proximal and distal portions of the PCL (**C**,**D**) MRI sagittal images of ramp and bucket handle lesions.

**Table 1 healthcare-12-01553-t001:** The demographic data.

Control Group (N = 76)	Study Group (N = 78)	
Age (y), mean ± SD	27.2 ± 8	26.4 ± 6.5
Sex		
Male	67 (86%)	61 (80%)
Female	11 (14%)	15 (20%)
Side		
Right	42 (54%)	45 (59%)
Left	36 (46%)	31 (41%)
Dominance		
Right	54 (69%)	55 (72%)
Left	24 (31%)	23 (28%)

Data are presented as mean ± standard deviation and number (percentage). Y, years; SD, standard deviation.

**Table 2 healthcare-12-01553-t002:** Additional information on the study group.

Associated Lesions	
Ramp lesion	37 (47%)
Bucket handle tear	9 (11%)
Lesion of the posterior horn of the medial meniscus	13 (17%)
Lachman Test	
Grade 1	23 (29%)
Grade 2	44 (56%)
Grade 3	11 (14%)
Pivot Shift Test	
Grade 1	23 (29%)
Grade 2	33 (42%)
Grade 3	22 (28%)
Delay (days)	333.4 ± 23.5

Data are presented as number (percentage).

**Table 3 healthcare-12-01553-t003:** Inter- and intraobserver reliability of posterior cruciate ligament (PCL) angle and tibial slope.

	Mean ± SD (°)		*p* Value	Inter-Observer Reliability	Intra-Observer Reliability
	Clinician 1	Clinician 2		ICC (95% CI)	ICC (95% CI)
PCL Angle					
Study group	110.7 ± 15.2	115.3 ± 16.2	<0.0001 * ^§^	0.866	0.955
Control group	111.4 ± 12	114 ± 14.5	0.0035 * ^§^	0.894	0.943
*p* value	0.991 ^ç^	0.421 ^ç^			
Tibial Slope					
Study group	9.3 ± 2.5	9.3 ± 2.8	0.683 ^§^	0.787	0.669
Control group	8.2 ± 1.8	7.7 ± 2.3	0.01 * ^§^	0.849	0.698
*p* value	0.007 * ^ç^	0.001 * ^ç^			

SD, standard deviation; ICC, intraclass correlation coefficient. * Statistical significance (*p* < 0.05). ^§^ Paired *t*-test. ^ç^ Unpaired *t*-test.

**Table 4 healthcare-12-01553-t004:** Univariate analysis.

Variables	*p*-Value
Grade of Lachman test	0.718
Grade of pivot shift test	0.449
Tibial slope	0.378
Ramp lesion	0.047 *
Bucket handle tear	0.004 *
Posterior horn of the medial meniscus tear	0.224
No meniscus tear	0.584
Delay injury-–urgery	0.138

* Statistically significant (*p* < 0.05).

**Table 5 healthcare-12-01553-t005:** Multivariate linear regression analysis.

Variables	Beta	95% CI		*p*-Value
		Lower Limit	Upper Limit	
Ramp Lesion	9.35	2.32	16.39	0.024 *
Bucket Handle Tear	−10.55	−19.69	−1.41	0.010 *

CI, confidence interval. * Statistically significant (*p* < 0.05).

## Data Availability

Data are contained within the article.

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
