# Peer review of "Posterior Cruciate Buckling Angle Variations Are Associated with Different Patterns of Medial Meniscus Tears in Anterior-Cruciate-Deficient Knees: Results of a Prospective Comparative Magnetic Imaging Resonance Study"

_healthcare, 2024, doi:10.3390/healthcare12161553_

Round 1
Reviewer 1 Report
Comments and Suggestions for Authors
I consider this paper a relevant issue in the field of sports medicine, particularly in the diagnosis of ACL tears and their relationship with PCL buckling angles and medial meniscus tears.
Line 26 : Cohort study. Method > Cohort study, Method
Lines 56-57 : “Diagnosis of ACL tear on MRI is usually based on direct and indirect radiographic 56 signs.”
I think it would be better to use radiologic signs instead radiographic signs.
Lines 79-80 : “ The delay between injury and surgery was recorded and its influence on the PCL angle was assessed.”
I invite you to better explain how the delay between injury and surgery can influence the PCL angle. Furthemore, you should show the values of the delay in each group. The delay between injury and surgery was recorded, but its specific impact on the PCL angle was not deeply analyzed in the results section.
References : I note a huge problem with yours references. For example : “ All patients were clinically evaluated in the surgical ward to surgery by a senior surgeon trained in sports medicine. The degree of anterior-posterior instability was recorded with Lachmann testing and graded according to the original description by Adler et al [12] with Grade 1 (mild): 3-5 mm translation of the tibia on the femur, Grade 2 (moderate): 5-10 mm translation of the tibia on the femur and Grade 3 (severe): >10 mm translation of the tibia on the femur. Rotatory instability was evaluated with the pivot shift test. It was graded according to the classification by Jacob et al [13] “ .
In your references Adler at al is [13] instead [12]. Similarly, Jacob et al is [14] instead [13]. You must review all your references.
Lines 92-95 : “ The integrity of the MCL and LCL was confirmed with the varus and valgus stress test. MRI as well as X-rays (including weight bearing lateral view with the knee at 15° of flexion) of the involved knee were available for all patients evaluated within 20 from surgery” .
You should specify what 20 refers to.
Table 1 : 333.4±23.5 > 333,4±23.5
Lines 184-187 : “The most important finding of the present study was the statistically significant association that emerged between bucket handle tears of the medial meniscus and lower PCL buckling angles (p=0.010) and between ramp lesions of the medial meniscus (p=0,024) 186 and greater PCL buckling angles. “
I totally agree with you regarding the most important findings of your study, but I think it would be noteworthy to portrait the values of PCL buckling angle with ramp lesion and bucket handle tear.
Line 205 : In the present ( without dot)
Lines 226-229 : “ This may be due to the dislocation of the meniscus in the intercondylar notch which may contact the PCL causing PCL stretching and increasing the PCL buckling angle and to the slight knee flexion due to the bucket handle tear and muscle contraction due to the pain.”
It’s widely known and accepted (DOI : 10.7759/cureus.43324 ) that one of the most common signs of bucket handle tears is the “ double PCL sign”. What was your strategy to avoid measurement errors?
Reviewer 2 Report
Comments and Suggestions for Authors
The topic of PCL buckling in relation to ACL tear has been studied widely, with varying conclusions. The rationale for studying this needs be strengthened (major concern): why various angles were measured and what they mean in relation to ACL tear is not explained. Another moderate weakness is the lack of good control group. Methodology is sparse (another major concern), although statistical analyses seem rigorous. Due to significant differences the measurements between the two readers, the confidence regarding the validity of the conclusion is low. Overall the paper feels rushed with numerous and inconsistent errors.
Introduction:
- Please combine scattered sentences into paragraphs with common topics.
- ACL tear can be directly imaged on MRI and this should be explained?
- clinical value of assessing PCL buckling could be better explained. Yes, past studies suggested association with ACL tear, but is it necessary to infer from PCL buckling when it is possible to directly evaluate ACL itself?
- The goal of the study is to show that PCL buckling is unreliable, but the mechanism or rationale for the hypothesis is not provided.
- The sentence "The purpose of the present study was to assess the variations of PCL buckling angle measurement is poor and cannot be used as a reliable tool in the MRI diagnosis of ACL tearing." does not make sense grammatically."
- importance of various anatomic factors needs to be better explained in Introduction.
Methods
- line 20: "within 20" days??
- For each type of measurement, provide brief methodology even if it has been previously published.
- what scanning parameters were used for knee MRI and on which images were the measurements made?
- the "Control" group is somewhat flawed since it has meniscal tear, which the ACL tear group does not have. How this affects the measurements is unclear.
- Please show in a figure how ramp lesions and bucket handle tears are detected.
- Lachmann should be Lachman
- Statistical methods appear rigorous.
Results
- please add p-values for Table 1 where applicable.
- Table 1: Lackmann should be Lachman
- Table 1: Shouldn't results for Lachman and Pivot test be shown under Study group column?
- there could be a figure comparing representative X-ray and MRI images from each cohort.
- Table 2: Significant differences between two readers suggests unreliability and possible bias (Clinician 2 gave greater PCL angle). This is concerning and lowers the confidence in the results and conclusion. Exact methodology used to measure this needs to be provided and scrutinized.
- Table 2: a low intra-observer reliability (lower than inter-observer) for tibial slope measurement is very odd. Exact methodology used to measure this needs to be provided and scrutinized.
Conclusions
- based on the weaknesses of the study, it would be difficult to state with confidence that ACL tear has no association with PCL buckling angle.
Comments on the Quality of English Language
several mistakes found.
Reviewer 3 Report
Comments and Suggestions for Authors
Thanks for the opportunity to review the current manuscript. The authors present a study in which they assesses the variations of PCL buckling angle in patients with ACL tears and in patients with isolated lesions posterior horn of the medial meniscus.
The research question is of relevant academic and clinical interest, as is the hypothesis that associated anatomic factors like posterior tibial slope and concomitant different patterns of medial meniscus tears might influence PCL buckling angle.
Regarding the first sentence: Do the authors not agree that the diagnosis of an ACL tear or insufficiency could also be made during a diagnostic arthroscopy? This applies to an even greater extent to some of the mentioned meniscus pathologies, as the authors correctly state. (lines 18-19)
This is particularly relevant as the control group consists of 76 patients undergoing a knee arthroscopy for isolated tears of the posterior horn of the medial meniscus and intact ACL. Was this checked again intraoperatively? Was the diagnosis always correct?
Especially since isolated tears of the posterior horn of the medial meniscus are a rarity in young patients with an intact and sufficient ACL and are instead commonly associated with insufficient ACLs.
Has plateau edema been evaluated in both groups? Especially for classic pivot-shift patterns and posteromedial tibial bone bruise?
The unusually long mean delay of 333.4±23.5 days should also be noted here, which presumably makes it even more difficult to draw a clear distinction between the two groups.
What exactly does the information in Table 1 mean, that in the control group, which was operated on due to tears in the medial posterior horn of the meniscus, there was ultimately no meniscus tear in 3%? Were these then excluded? Or were they included in the calculations?
Please specify how and to what extent the two independent reviewers were blinded.
Other statements relating to the examination of the ACL are also simplified and do not do justice to the usual clinical and radiological examination steps. (e.g. line 51)
Please write out small numbers, e.g. line 111.
Would it be possible to add Figure 2 without the drawn angles to allow a meaningful understanding of the performed measurement as orinally described by Yoon et al.
In view of these multiple methodological issues, the limitations are surprisingly brief. Conclusions such as that "the PCL buckling angle did not show statistically significant difference in patients with or without ACL tears. Therefore, the angle should not be used as an indirect radiographic sign of ACL tearing." should therefore be viewed with considerable caution.
Please further adjust
"Abstract :Background:" (line 18)
"fractrures" (line 82)
"within 20 from surgery." (line 95)
"Partecipant characteristic" (line 141)
"In the present." (line 205)
"Lackmann" (Table 1)
Comments on the Quality of English Language
Moderate editing of English language required.
Reviewer 4 Report
Comments and Suggestions for Authors
You have conducted a very important, interesting and well done reseach!
Only minor suggestions.
The sentence in lines 67-68 should be changes to the more understandable.
Line 71 - it is better to change "measure" to "measurement" , in my opinion.
It would be better to describe, what do you mean saying "greater" or "lower" buckling angles.
Greater angle or greater buckling? I think I understand, but it is better not to have doubts about it.
Round 2
Reviewer 1 Report
Comments and Suggestions for Authors
The paper has been considerably improved
Author Response
Thank you very much.
Reviewer 2 Report
Comments and Suggestions for Authors
General comment:
It is difficult to tell where and what changes have been made without comparing old vs. new manuscript side by side. Please use proper marking tools to make the changes stand out.
Comments on:
Response 3 (R3): It is important to have good rationale for any study. If PCL buckling has no meaning, why was it studied by many radiologists in the past? Please provide more Introduction on the meaning of PCL buckling for knee health.
R8: I see. Please add this to the Methods under patient description. It is not clear.
R15: Since there are two cohorts, showing images from each cohort would be useful (showing ACL tear and PCL buckling for example).
R16: There needs to be further analysis or consideration of the effect of reader bias on the measurement.
R17: Tibial slope measurement on x-ray should be even more reproducible than buckling angle measurement. ICC less than 0.7 tells me that that the measurements were not performed with care and standard protocol.
R18: There are two issues here. 1) the reading of PCL angle from two readers is different. How this affects the conclusion was not considered. 2) Ignoring the reader-bias, the statistics shows that there is no difference between groups in the PCL angle. This does not conclusively mean that there is an absence of an association. Either perform additional analyses to show that, or revise the conclusion.
